# The Effect of Cold Stress on the Root-Specific Lipidome of Two Wheat Varieties with Contrasting Cold Tolerance

**DOI:** 10.3390/plants11101364

**Published:** 2022-05-20

**Authors:** Bo Eng Cheong, Dingyi Yu, Federico Martinez-Seidel, William Wing Ho Ho, Thusitha W. T. Rupasinghe, Rudy Dolferus, Ute Roessner

**Affiliations:** 1Biotechnology Research Institute, Universiti Malaysia Sabah, Jalan Universiti, Kota Kinabalu 88400, Malaysia; 2School of Bio Sciences, The University of Melbourne, Parkville, VIC 3010, Australia; dyu@svi.edu.au (D.Y.); fmartinezsei@student.unimelb.edu.au (F.M.-S.); u.roessner@unimelb.edu.au (U.R.); 3Protein Chemistry and Metabolism Unit, St. Vincent’s Institute of Medical Research, Fitzroy, VIC 3065, Australia; 4Max-Planck-Institute of Molecular Plant Physiology, Am Mühlenberg 1, 14476 Potsdam, Germany; 5Advanced Genomics Division, Walter and Eliza Hall Institute of Medical Research, Parkville, VIC 3052, Australia; wing.ho@unimelb.edu.au; 6SCIEX, Mulgrave, VIC 3170, Australia; thusi.rupasinghe@sciex.com; 7CSIRO Agriculture & Food, GPO Box 1700, Canberra, ACT 2601, Australia; rdolferus58@gmail.com; 8Research School of Biology, The Australian National University, Acton, ACT 2601, Australia

**Keywords:** chilling, freezing, lc-tandem-ms, lipidome, spatial root, wheat, young, wyalkatchem

## Abstract

Complex glycerolipidome analysis of wheat upon low temperature stress has been reported for above-ground tissues only. There are no reports on the effects of cold stress on the root lipidome nor on tissue-specific responses of cold stress wheat roots. This study aims to investigate the changes of lipid profiles in the different developmental zones of the seedling roots of two wheat varieties with contrasting cold tolerance exposed to chilling and freezing temperatures. We analyzed 273 lipid species derived from 21 lipid classes using a targeted profiling approach based on MS/MS data acquired from schedule parallel reaction monitoring assays. For both the tolerant Young and sensitive Wyalkatchem species, cold stress increased the phosphatidylcholine and phosphatidylethanolamine compositions, but decreased the monohexosyl ceramide compositions in the root zones. We show that the difference between the two varieties with contrasting cold tolerance could be attributed to the change in the individual lipid species, rather than the fluctuation of the whole lipid classes. The outcomes gained from this study may advance our understanding of the mechanisms of wheat adaptation to cold and contribute to wheat breeding for the improvement of cold-tolerance.

## 1. Introduction

Wheat (*Triticum aestivum* L.) is the most widely cultivated crop on Earth, yielding ~750 million tons in 2016 for human nutrition. The production of wheat needs to increase by 60% to feed the projected world population of 9.6 billion by the year 2050 (International Wheat Genome Sequencing Consortium, www.wheatgenome.org; date accessed 7 April 2022). A key challenge for wheat breeding is the development of varieties with tolerance to multiple biotic and abiotic stresses, while maintaining yield stability [1,2].

Low temperatures, such as chilling (0–12 °C) and freezing (<0 °C), are one of the major yield-limitations to wheat productivity. The capacity to withstand cold temperatures in wheat may vary depending on the genetic background and variability. Some wheat varieties are cold tolerant, while others are more sensitive. Nonetheless, all these varieties need to go through a similar adaptation process during the exposure to low, non-freezing temperatures to acquire freezing tolerance. This process is referred to as cold acclimation [3,4].

Induction of cold acclimation and freezing tolerance involves physiological and biochemical changes in plants [5]. Plasma membranes are the primary barrier between the organism and the external environment, being the first to experience the injurious effects of stress. The plasma membrane transitions from a fluid state to a rigid gel phase when exposed to low temperatures. In order to maintain the fluidity and re-stabilize the membrane at low temperatures, the cell increases the level of unsaturated lipids, alters the chain length and lipid class composition, and changes the membrane lipid-to-protein ratio [6]. All these mechanisms lead to the remodeling of the membrane, which allows the cell to mechanically adapt to cold [6,7,8]. In the cell, not only does the plasma membrane undergo such remodeling, but the endomembrane of other organelles, such as the chloroplast, the endoplasmic reticulum, the Golgi apparatus, and the tonoplast, also undergo similar remodeling when exposed to low temperatures [6,8].

Plant membranes (both plasma membrane and endomembrane) are composed of three main categories of lipids: (1) glycerolipids such as phosphoglycerolipids, galactosyl glycerolipids, and di/triacylglycerols, (2) sphingolipids, and (3) sterols. Glycerolipids that contain phosphate groups in their structures are called phosphoglycerolipids. For example, the phosphotidylcholines (PCs) and phophotidylethanoamines (PEs) are the most abundant glycerolipids present in the extra plastidial membranes, such as the plasma membrane [9]. Glycerolipids that contain galactose in their structures are called galactosyl glycerolipids. Galactosyl glycerolipids are present predominantly in the plastidial membranes, such as the chloroplastic thylakoid membranes. The thylakoid membrane is composed of 50% monogalactosyldiacylglycerol lipids (MGDG), 26% digalactosyldiacylglycerol lipids (DGDG), and the remaining lipids are sulfoquinovosyldiacylglycerol (SQDG) and phosphatidylglycerol lipids (PGs) [10]. There are also glycerolipids that do not contain any phosphate or sugar groups in their structures, such as diacylglycerols (DGs) and triacylglycerols (TG). DGs are mainly used for membrane lipid assembly during vegetative growth to support cellular membrane biogenesis, expansion, and maintenance [11]. Different from the category of plant glycerolipids, plant sphingolipids are structurally diverse molecules composed of three building blocks: a sphingoid long-chain base (LCB) backbone amide-linked to a fatty acid (FA), which is bonded to a polar head group [12]. Sphingolipids compose an estimated 40% of plasma membrane lipids and are enriched in the outer membrane layer where they influence membrane integrity and ion permeability [13,14,15]. The third category of plant lipids, the sterol lipids, fill the voids between sphingolipids to increase the packing of lipids, and together with specific phosphoglycerolipids, they form membrane domains, such as lipid rafts [9,16,17]. Besides the structural protection of the cell membranes and involvement in remodeling during cold stress, plant lipids also serve as signaling molecules when plants are exposed to low temperatures. Lipid signaling molecules are in low abundance (less than 1% of total lipids), accumulate transiently, and have a fast turnover rate [18]. Signaling lipids include phosphatidic acid (PA), phosphoinositides (PIs), sphingolipids, lysophospholipids, diacylglycerols (DGs), oxylipins, and others [19,20,21,22,23,24,25].

Considering these important roles of different lipid species in plants upon low temperature stress, it is essential to investigate their profiles and changes in plants subjected to cold stress to understand the underlying mechanism in depth. The majority of studies on lipidome changes in plants exposed to cold stress were performed in only a small number of species other than wheat, such as *Arabidopsis thaliana* [7,8,21,26,27,28,29,30], rice [31], maize [32,33], and barley [34,35].

Several studies have reported on the lipid profiling of a particular lipid class or more complicated lipidome analyses upon cold stress in wheat [36,37,38,39,40,41,42,43]. In [36], Roche et al. studied the phospholipid content of seedlings grown at 2 °C and reported that the content was higher than that of seedlings grown at 24 °C. Ashworth et al. [37] reported the decrease of linoleic acid but increase in linolenic acid in wheat seedling roots in response to declined temperature, while phospholipid was not influenced by temperature. Vigh et al. [38] studied the correlation of glycolipids and fatty acids of chloroplast thylakoid membranes in two wheat varieties of contrasting hardiness. Kendall et al. [39] isolated the crowns from seedlings of *Triticum aestivum*, L. cultivar Fredrick and reported an increased lipid phase transition temperature, loss of lipid phosphate (lipid-P), and increased free fatty acid levels after a lethal freeze-thaw stress. In [40], Bohn et al., reported an increase of sterols and reduction of glycolipids in the plasma membranes isolated from acclimated wheat seedlings. Li et al. [41] reported that cerebroside C increased tolerance to chilling injury and altered lipid composition in the roots of wheat seedlings. In all these studies, the tissues used for the analyses were either shoots, seedlings, or whole seedling roots [36,37,38,39,40,41]. However, a thorough investigation of the effect of cold stress in roots or more specifically in the different developmental root zones of the young roots has not been conducted. Our targeted lipidomics study on the aerial part (flag leaf [42] and spike [43]) of two wheat varieties with contrasting cold tolerance upon cold stress has revealed the potential involvement of certain phosphoglycerolipids in membrane re-modelling in cold stress, which differentiate the two wheat varieties. Although the roots are less exposed to cold than the aerial plant parts, cold temperatures are likely to affect root development as well, and signals from cold-stressed above-ground plant parts will affect root growth. It is therefore interesting to further study the lipid profiles of the roots of the two wheat varieties, especially on the different development root zones. Sarabia et al. [44,45] and Ho et al. [46] have recently reported comprehensive comparative spatial lipidomics analyses on different developmental root zones of barley subjected to salt stress and revealed cellular lipid remodeling in different developmental zones of barley roots in response to salinity. These results demonstrate the differential impact of salinity on different developmental root zones in each studied cultivar, as well as between different barley cultivars with differing levels of salinity tolerance. This encouraged us to apply a similar approach to investigate the impact of cold stress on the developmental zones of roots of two wheat cultivars with differing tolerance levels to cold stress.

Roots are vital to anchoring the plant in soil and taking up water and nutrients. Nielsen and Humphries [47] have reported that root growth is generally more sensitive to temperature than that of above-ground plant parts, and root growth may cease altogether if soil temperatures drop below 2 °C [48]. In the tip of a root, there are different developmental zones along the root’s longitudinal axes: the root cap and meristemic (cell division) zone, which contains the active dividing cells, the elongation zone in which cells rapidly expand along the longitudinal axis, and the differentiation (maturation) zone in which cells differentiate to become functionally specialized [49] (Figure 1A). In this study, we aimed to investigate the lipid profiles across the spatial root zones of two cold-stressed spring wheat varieties with contrasting cold-tolerance, with Wyalkatchem being cold-sensitive and Young being cold-tolerant [42,43], using a comprehensive targeted lipidomics approach developed by Yu et al. [50,51]. Two-day-old, germinated seedlings of the two wheat varieties were subjected to optimal (21 °C), chilling (4 °C), and freezing (−3 °C) temperatures for six hours. The meristemic/division (DZ), elongation (EZ), and maturation (MZ) zones of the roots were harvested, and the lipids were extracted and analyzed. We profiled 273 lipid species derived from 21 lipid classes, which revealed distinct lipidomes of the different root zones in each variety, as well as between the two varieties in response to chilling and freezing stresses. Together with the studies reported by Cheong et al. [42,43] on the flag leaf and spike lipidome, the outcomes gained from this study (roots) may help us to better understand the cold response mechanism of wheat, thus contributing to the development of wheat breeding programs and the improvement of more cold-tolerant varieties in the future.

## 2. Results

### 2.1. Targeted Profiling Using Scheduled Parallel Reaction Monitoring Assays Enables the Profiling of the Complex Lipidome in Different Wheat Root Developmental Zones

Using the scheduled parallel reaction monitoring (PRM) assays, a total of 273 lipid species derived from 21 lipid classes that fall under the three lipid categories (glycerolipids, sphingolipids, and sterols) were detected, measured, and analyzed in the three wheat root developmental zones (Figure 1). All 273 lipid species analyzed for each class are summarized in Appendix A. Among the 21 classes of lipids detected and summarized in Figure 1, the sphingolipid monohexosyl ceramides (HexCer) contributed to the highest number of detected lipid species (41 species). This was followed by the glycerolipids, such as phosphatidylethanolamines PE (36 species), diacylglycerols DG (31 species), and phosphatidylcholines PC (22 species). The PRM assays also enabled the detection and measurement of ceramides Cer (21 species), phophatidylglycerol PG (20 species), phosphatidylinositide PI (12 species), digalactosyl diacylglycerols DGDG (12 species), sulfoquinovosyl diacylglycerol SQDG (11 species), and sterols such as acylated steryl glycoside ASG (10 species).

### 2.2. Overview of the Lipid Profiles in Each of the Developmental Root Zones of Wyalkatchem and Young Varieties at the Unstressed Stage, and after Chilling and Freezing Stress

First of all, it is important to understand the distribution of the measured 273 lipids from the 21 lipid classes in the three developmental root zones of the two wheat varieties with contrasting cold tolerances prior to any cold treatment (unstressed stage). We have therefore calculated the proportion of each lipid class as percent of mass spectral intensity. Figure 2 shows the distribution of the 21 lipid classes in each specific root zone of cold-sensitive Wyalkatchem and cold-tolerant Young at the unstressed stage.

At the unstressed stage, in the division zone (DZ) where cells are actively dividing, it was observed that phosphatidylethanolamine (PE) was the most abundant lipid class in Wyalkatchem (29.4%), followed by monohexosyl ceramides (HexCer) with 22.8%. In contrast, in Young, HexCer occupied the highest abundance in the DZ (31.6%), followed by PE with 22.8% (Figure 2). Phosphatidylglycerol (PG) was the third most abundant lipid class for Wyalkatchem (15.9%), followed by phosphatidylcholines (PC, 10.6%), diacylglycerols (DG, 10.2%), lysophospholipids, and other classes of lipids. This lipid distribution was slightly different in Young, where DG (10.8%) was the third most abundant lipid class in this root zone (DZ), followed by PG (10%), PC (8.1%), and other classes of lipids (Figure 2). In the elongation zone (EZ), where cells expand in size, both wheat varieties showed the same patterns of lipid distribution. It was observed that HexCer was the most abundant class in both Wyalkatchem (34.8%) and Young (42.1%). This was followed by PE (25.8% in Wyalkatchem and 21.0% in Young), DG (10.7% in Wyalkatchem and 10.2% in Young), PG (10.3% in Wyalkatchem and 7.0% in Young), PC (9.6% in Wyalkatchem and 7.0% in Young), and the remaining classes. In the maturation zone (MZ), where cells start to differentiate for specialized functions, it was observed that the lipid distributions in both the wheat varieties were similar. HexCer was the most abundant lipid class in both the wheat varieties with 43.3% in Wyalkatchem and 45.4% in Young, followed by PE (21.3% in Wyalkatchem and 20.1% in Young), DG (12.1% in Wyalkatchem and 10.3% in Young), PG in Wyalkatchem (7.5%) in Wyalkatchem but PC in Young (6.2%), then PC in Wyalkatchem (7.2%), and PG in Young (6.2%), and the remaining classes of lipids (Figure 2). Overall, the Young variety has higher abundances of sphingolipids (Cer and HexCer) compared to Wyalkatchem in all three root zones, while Wyalkatchem has higher abundances of phosphoglycerolipids (PE, PG, PE, PI, and PS) compared to Young in the three root zones. Young also has higher abundances of lysophospholipids compared to Wyalkatchem in the three root zones. There were not many differences of DG between the two varieties in the DZ and EZ, but levels of this lipid class were higher in Wyalkatchem compared to Young in the MZ (Figure 2).

When comparing the compositions of the measured lipids “across” the three root zones (from DZ to EZ to MZ) in each variety, it was interesting to observe that the percentages for all phosphoglycerolipids (PE, PG, PC, PI, and their lyso species) decreased in the EZ compared to the DZ for both wheat varieties. The EZ of Wyalkatchem contained 47.6% phosphoglycerolipids compared to its DZ (59.9%), while the EZ of Young contained 40.3% phosphoglycerolipids compared to its DZ (50.9%). The phosphoglycerolipids were lower in the MZ of both varieties compared to the EZ or DZ (37.5% and 35.6% in Wyalkatchem and Young, respectively; Figure 2). Interestingly, percentages for sphingolipids such as Cer and HexCer were higher in the EZ compared to the DZ in both varieties. The EZ of Wyalkatchem contained 37.0% sphingolipids compared to 24.5% in DZ, while the EZ of Young contained 43.8% sphingolipids compared to 32.9% in DZ. The sphingolipids were further increased in the MZ compared to EZ or DZ for both varieties: 45.9% and 47.9% in Wyalkatchem and Young, respectively. The DG was observed to be higher in the MZ (12.1%) of Wyalkatchem roots compared to the DZ (10.2%) and the EZ (10.7%), while there was no obvious difference of this lipid class amongst the three root zones in Young (Figure 2). The sterol ASGs were observed to be slightly higher in the MZ of Young compared to its DZ and EZ, while there was no difference for this class of lipids in Wyalkatchem. Moreover, there were not many differences of galactosyl glycerolipids among the three root zones at the unstressed stage for both wheat varieties (Figure 2).

After six hours of exposure to chilling and freezing stress, several interesting changes in the lipids were observed. The first distinctive change was the increase of phosphoglycerolipids (PG, PE, and PC) in all three root zones of Wyalkatchem (Figure 2A) and Young (Figure 2B). The second distinctive change in lipid levels was observed for the sphingolipids HexCer and Cer classes. Chilling decreased these classes of lipids in all three root zones of both the varieties, but freezing only decreased these classes of lipids in the DZ and EZ of both the varieties with a very slight decrease in MZ (Figure 2). The third main change in lipids happened for the DG class. Both six hours of chilling and freezing exposures caused an increase of this class of lipids in the EZ and MZ of Wyalkatchem and Young, but there was no effect in the DZ (Figure 2).

### 2.3. Varietal Difference: Comparisons of Lipid Profiles of Each of the Developmental Root Zones of Cold-Tolerant Young with Cold-Sensitive Wyalkatchem after Chilling and Freezing Stress

To study the differences in lipid profiles between the cold-tolerant and cold-sensitive varieties in response to cold stress, we compared the corresponding lipid profiles of the root zones between cold-tolerant Young and cold-sensitive Wyalkatchem following chilling and freezing stress (Figure 3). The lipid profiles for all three developmental root zones between Wyalkatchem and Young revealed significant differences in phosphoglycerolipids, such as lysophospholipids (lyso PC, lysoPE, lysoPG, lysoPS, lysoPI), PC, PE, PG, PI, PS, and galactosyl glycerolipids, such as DGDG, DGMG, MGDG, MGMG, SQDG, and SQMG. Lipid profiles of DG, Cer, and HexCer species were also different for the two varieties, but they were not as obviously different compared to the phosphoglycerolipids and galactosyl glycerolipids (Figure 3, Appendix A).

In the DZ, it was observed that the majority of the lysophospholipids (phospholipids with single acyl chain in their structures), such as lysoPC, lysoPE, lysoPG, lysoPI, and lysoPS species, were significantly higher in cold-tolerant Young compared to cold-sensitive Wyalkatchem after the cold exposures. In the DZ, 23 out of 25 and all 25 lysophospholipids were higher in Young compared to Wyalkatchem following both chilling and freezing stress (Figure 3, Appendix A). The opposite was true for the phospholipids with two acyl chains, in which the majority of the lipids were lower in Young compared to Wyalkatchem. For example, 24 and 26 PC species were significantly lower in chilling and freezing stressed Young compared to Wyalkatche, respectively (Figure 3). The same was observed for PE, in which 21 and 20 species were significantly lower in Young compared to Wyalkatchem, respectively (Figure 3). For other phospholipids such as PG, PI, and PS, the majority of the species were also significantly lower in Young compared to Wyalkatchem (Figure 3). For the galactosyl glycerolipid profiles, it was interesting to observe that the majority of the species that contained diacylglycerols in their structures, such as DGDG (digalactosyl diacylglycerol), MGDG (monogalactosyl diacylglycerol), and SQDG (sulfoquinovosyl diacylglycerol), were significantly lower in Young compared to Wyalkatchem after cold stress (Figure 3, Appendix A). Conversely, species that contained monoacylglycerol in their structures, such as DGMG (digalactosyl monoacylglycerol), MGMG (monogalactosyl monoacylglycerol), and SQMG (sulfoquinovosyl monoacylglycerol) were significantly higher in Young compared to Wyalkatchem after cold stress (Figure 3 and Appendix A). For sphingolipids (Cer and HexCer) and sterols (ASG and SG), there were not many significant differences in this root zone between Young and Wyalkatchem. Only five Cer and four HexCer species were observed to be lower after being subjected to freezing stress in Young compared to Wyalkatchem (Figure 3 and Appendix A). All the fold changes of the lipids that were significantly different between Young and Wyalkatchem for the DZ are listed in Appendix A.

The patterns of phosphoglycerolipids and galactosyl lipids observed in the DZ were also found in the EZ. The majority of the lysophospholipids were significantly higher in Young compared to Wyalkatchem in the EZ after cold stress (Figure 3, Appendix A). Meanwhile, the majority of the phospholipids (PC, PE, PG, PI, and PS) were significantly lower in Young compared to Wyalkatchem after cold stress. Interestingly, the numbers of PC and PE (the two most abundant phospholipid classes in plant membranes) that were significantly lower in Young compared to Wyalkatchem were higher in the EZ (Figure 3). Galactosyl glycerolipids such as DGDG, MGDG, and SQDG were again lower in Young compared to Wyalkatchem in this root zone under cold-stressed conditions. Meanwhile, DGMG, MGMG, and SQMG were again significantly higher in Young compared to Wyalkatchem in this root zone under both cold-stressed conditions (Figure 3 and Appendix A). It was interesting to observe significant differences of DG, Cer, and Hexcer species between the two varieties in this root zone, after being subjected to chilling stress. This phenomenon was not observed in the DZ. Twenty-one DG, eight Cer, and 15 HexCer were significantly lower in Young compared to Wyalkatchem after chilling stress. However, different from chilling stress, there were no significant differences of DG, Cer, and HexCer in the EZ after the freezing stress (Figure 3, Appendix A). All the fold changes of the lipids that were significantly different between Young and Wyalkatchem in the EZ are listed in Appendix A.

In the MZ, the majority of lysophospholipids (PC, PE, PG, PI, and PS) were significantly higher in Young compared to Wyalkatchem after exposure to chilling and freezing stress (Figure 3, Appendix A). For phospholipids, the numbers of PC, PE, and PG species that were significantly different between Young and Wyalkatchem increased after chilling and freezing stress. For example, 24 PC, 18 PE, and 11 PG species were lower in Young compared to Wyalkatchem after chilling stress. Meanwhile, 24 PC, 15 PE, and six PG species were lower in Young compared to Wyalkatchem after freezing stress (Figure 3 and Appendix A). For the galactosyl glycerolipids, DGDG, MGDG, and SQDG were again lower in the MZ of Young compared to Wyalkatchem after chilling and freezing stress, as was also the case for the other two root zones. Meanwhile, similar to the other two root zones, DGMG, MGMG, and SQMG were significantly higher in Young compared to Wyalkatchem in the MZ following freezing stress (Figure 3 and Appendix A). Similar also to what was observed following chilling stress, lipid species from the DG, Cer, and HexCer classes were significantly lower in Young compared to Wyalkatchem (Figure 3 and Appendix A). All the fold changes of the lipids that were significantly different between Young and Wyalkatchem in MZ are listed in Appendix A.

### 2.4. Lipid Species Contributing Changes in Lipid Profiles in the Three Developmental Root Zones of Both Wheat Varieties following Cold Stress

Considering the effects of cold treatment on phosphoglycerolipids, galactosyl lipids, and sphingolipids for the three different developmental root zones in Wyalkatchem and Young, it was important to know which specific lipid species contributed most to these effects in each root zone for each variety. Hierarchical clustering analysis (HCA) was performed on all the root samples (control and cold stress, three developmental root zones) of the two wheat varieties (Figure 4). As shown in Figure 4, a total of seven clusters of lipid species were clustered based on the wheat varieties, root samples, and types of cold stress (control, chilling, and freezing). Several clusters with interesting lipids were observed. For example, the lipid species that were present in the DZ of cold-tolerant Young at the unstressed and cold-stressed states are grouped in Cluster 3 (Figure 4; marked with rex box). The lipid species included 25 lysophospholipid species, such as LysoPE(18:2), LysoPE(22:1), LysoPC(16:0), LysoPC(16:1), LysoPS(18:2), LysoPI(18:2), LysoPI(18:3), LysoPG(18:3), LysoPG(16:0), LysoPG(18:2), and LysoPG(18:1). This cluster also contains 10 monoacylglycerol species, such as SQMG(16:0), SQMG(18:2), MGMG(18:2), DGMG(18:2), and others as listed in Appendix A. Interestingly, these species were also significantly higher in cold-tolerant Young compared to cold-sensitive Wyalkatchem as revealed in the Log_2_-tranformed fold changes plot (Appendix A). Interesting as well was the group of lipid species present in Cluster 4 for the EZ of Wyalkatchem subjected to chilling and freezing stress (Figure 4; marked with rex box). The lipid species involved were 13 phosphoglycerolipids, such as PG(18:1_18:1), PG(18:0_18:2), PC(14:0_18:2), PE(16:0_18:1), PE(16:1_16:1), PS(16:1_18:2), and others (Appendix A); 12 sphingolipids, such as Cer(t18:1_24:0), HexCer(t18:1_23:1-OH), HexCer(d18:2_25:1-OH), HexCer(t18:1_25:1-OH), and others. The cluster also contains four galactosyl glycerol lipids, such as SQDG(14:0_18:3), SQDG(18:1_18:1), DGDG(18:1_18:3), and MGDG(16:1_16:1) (Appendix A). The third interesting cluster was Cluster 5 for the DZ of chilling and freezing stressed Wyalkatchem. This cluster contained the highest numbers of lipid species amongst the seven clusters, including 60 phosphoglycerolipids and 19 galactosyl glycerol lipids (summarized in Appendix A).

When the Log_2_ transformation of fold changes was performed for the main phosphoglycerolipids (PC, PE and PG), diacylglycerols (DG), and sphingolipids (Cer and HexCer) for the DZ, EZ, and MZ of Wyalkatchem and Young following chilling and freezing stress, more potential lipid species were observed. The significance of the changes in individual lipid species was determined using Student’s *t*-test with a false discovery rate (FDR)-adjusted *p*-value less than 0.05 as the cut-off (Benjamini-Hochberg, 1995). Lipid species that significantly increased after cold stress compared to control were colored red, while species that significantly decreased were colored green (Figure 5, Figure 6, Figure 7, Figure 8, Figure 9 and Figure 10). A large number of lipid species were found to be statistically significant, as shown in Figure 5, Figure 6, Figure 7, Figure 8, Figure 9 and Figure 10. Only species that changed equal to or more than −2.0-fold and +2.0-fold (while also statistically significant) are discussed in the results section.

Figure 5 shows the chilling and freezing effects on the DZ of Wyalkatchem and Young for lipid species of the HexCer, Cer, and DG classes. In Wyalkatchem, after exposure to chilling stress, DG(18:0_18:1) was the only DG species that decreased in the DZ, while DG(18:2_24:1) and DG(18:3_24:1) were both increased (Figure 5A). Compared to chilling, exposure to freezing stress caused more significant changes of DG species in the DZ of Wyalkatchem (Figure 5B). Four DGs were decreased in this zone, two of them more than −2.0-fold: DG(18:0_18:1) and DG(18:1_18:1). Five DGs were increased after freezing stress and four of these increased more than +2.0-fold: DG(18:2_24:0), DG(18:2_24:1), DG(18:3_24:0), and DG(18:3_24:1) (Figure 5B). After exposure to chilling stress, nine DG species decreased in the DZ of Young, three of which increased more than −2.0-fold: DG(16:0_16:0), DG(16:0_18:0), and DG(18:0_18:1) (Figure 5A). Compared to chilling, exposure to freezing stress caused less significant changes of the DG species in the DZ of Young (Figure 5B). Four DGs were observed to decrease in this zone, but all by less than −2.0-fold. Meanwhile, four DGs were observed to increase more than 2.0-fold after freezing stress: DG(18:2_24:0), DG(18:2_24:1), DG(18:3_24:0), and DG(18:3_24:1) (Figure 5B). Chilling and freezing also caused significant decreases of sphingolipids (Cer and HexCer) in the DZ of Wyalkatchem and Young (Figure 5). Chilling stress reduced four Cer species by more than −2.0-fold and no significant changes of HexCer species were found. The four Cer species were Cer(t18:0_22:0), Cer(t18:0_24:0), Cer(t18:0_24:1-OH), and Cer(t18:0_26:0) (Figure 5A). Compared to chilling stress, freezing caused an even stronger decrease of Cer species in the DZ of Wyalkatchem and seven species were involved (Figure 5B). Of these, six were decreased by more than −2.0-fold: Cer(t18:0_24:0), Cer(t18:0_24:1-OH), Cer(t18:0_25:0-OH), Cer(t18:0_25:1-OH), Cer(t18:0_26:0), and Cer(t18:0_26:1). One HexCer (HexCer(d18:2_18:1-OH) was decreased in the MZ following exposure to freezing stress, but the decrease was less than −2.0-fold (Figure 5B). Chilling caused many significant decreases of Cer (11) and HexCer species (21) in the DZ of Young (Figure 5A). Of the eleven Cer species, eight were decreased greater than −2.0-fold: Cer(18:0_22:0), Cer(t18:0_22:1-OH), Cer(t18:0_24:0), Cer(t18:0_24:1), Cer(t18:0_24:1-OH), Cer(t18:0_26:0), Cer(t18:0_26:1-OH), and Cer(t18:1_24:0). Of the 21 HexCer species, only six were decreased by more than −2.0-fold: HexCer(d18:1_16:0-OH), HexCer(d18:2_16:0-OH), HexCer(d18:2_18:0-OH), HexCer(d18:2_18:1-OH), HexCer(d18:2_20:1-OH), and HexCer(t18:1_18:0-OH) (Figure 5A). Freezing stress had less of an effect on Cer and HexCer compared to chilling in the DZ of Young (Figure 5B). Six Cer and five HexCer were significantly decreased following freezing stress. Of the six Cer species, five were decreased by more than −2.0-fold: Cer(t18:0_24:1), Cer(t18:0_24:1), Cer(t18:0_24:1-OH), Cer(t18:0_26:0), and Cer(t18:0_26:1). Two HexCer decreased by more than −2.0-fold: HexCer(t18:0_22:1-OH) and HexCer(t18:1_24:1-OH) (Figure 5B). Neither chilling nor freezing stress caused many significant changes of phosphoglycerolipids (PC, PE, and PG) in the DZ of both Wyalkatchem and Young (Figure 6). While chilling did not cause any significant changes to PC, PE, or PG levels in both varieties (Figure 6A), freezing caused a decrease of only one PE in Wyalkatchem (PE(16:0_16:1). In Young, freezing caused the decrease and increase of two of PE species: PE(14:0_18:2) and PE(14:0_18:3), respectively (Figure 6B). The increased PE species were PE(16:0_16:0) and PE(16:1_16:1). Freezing also caused the increase of two PG species in the DZ of Young: PG(16:0_16:0) and PG(16:1_16:1) (Figure 6B).

Next, the chilling and freezing effects on the elongation zone (EZ) of Wyalkatchem and Young for lipid species of the HexCer, Cer, and DG classes are shown in Figure 7, and those for lipid species of the PC, PE, and PG classes are shown in Figure 8. Chilling and freezing did not cause any significant changes in sphingolipids (Cer and HexCer) in the EZ of Wyalkatchem and Young (Figure 7). Chilling and freezing also did not cause any significant changes of DG in the EZ of Wyalkatchem and Young, except only two DG species that were significantly increased after chilling exposure in Wyalkatchem: DG(18:2_24:1) and DG(18:3_24:1) (Figure 7A). The most obvious effects of chilling and freezing on the EZ were observed in phosphoglycerolipids from the PC, PE, and PG classes, of which many of the lipid species were significantly increased in cold-sensitive Wyalkatchem after cold stress. Chilling caused the increase of 16 PC species in Wyalkatchem, five of which were increased more than +2.0-fold (Figure 8A). The five PC species were PC(18:3_18:3), PC(18:3_20:2), PC(18:3_22:1), PC(18:3_24:1), and PC(18:3_26:1). Chilling also induced the increase of 18 PE species in the EZ of Wyalkatchem, but only three of them were increased more than +2.0-fold: PE(18:3_18:3), PE(18:3_20:3), and PE(18:3_24:1). Nine PG species were increased in the EZ after chilling, but only one PG(18:3_18:3) was increased more than +2.0-fold (Figure 8A). Freezing caused even more significant changes to the PC, PE, and PG species in the EZ of Wyalkatchem compared to chilling (Figure 8B). A total of 24 PC species were significantly increased, with seven of them more than +2.0-fold. The PC species were PC(18:0_18:3), PC(18:3_18:3), PC(18:3_20:1), PC(18:3_20:2), PC(18:3_22:1), PC(18:3_24:1), and PC(18:3_26:1). Twenty-six PE species were increased, with ten of them by more than +2.0-fold: PE(18:0_18:3), PE(18:1_18:3), PE(18:2_18:3), PE(18:3_18:3), PE(18:3_20:1), PE(18:2_20:3), PE(18:3_22:0), PE(18:3_22:1), PE(18:3_24:0), and PE(18:3_24:1). One PE species was significantly decreased in the EZ of Wyalkatchem after freezing stress, i.e., PE (16:0_16:1). As for PG, ten species were increased in the EZ after freezing stress, but only two species by more than +2.0-fold: PG(18:2_18:3) and PG(18:3_18:3). One PG species was significantly decreased in this zone following freezing stress: PG(16:0_16:1) (Figure 8B). For the EZ of Young, chilling did not cause any significant changes to the PC, PE, and PG (Figure 8A), but freezing did have some significant effects (Figure 8B). Two PC species were significantly increased after freezing stress, but the changes were less than +2.0-fold. Four PE species were significantly increased and only two of them by more than +2.0-fold: PE(16:1_16:1) and PE(18:3_18:3). One PG species, PG(18:3_18:3), was significantly increased in this root zone by freezing stress, but by less than +2.0-fold change (Figure 8B).

For the Wyalkatchem MZ, chilling also caused a significant increase of two Cer, but by less than +2.0-fold, as well as one HexCer(d18:2_26:0) by +2.1-fold (Figure 9A). Compared to chilling stress, freezing stress again caused more increases of Cer and HexCer species in the MZ of Wyalkatchem (Figure 9B). Six Cer species increased by more than +2.0-fold: Cer(t18:0_22:1), Cer(t18:1_22:0), Cer(t18:1_22:0-OH), Cer(t18:1_24:0), Cer(t18:1_24:1-OH), and Cer(t18:1_26:0). As opposed to Wyalkatchem, both chilling and freezing did not have significant effects on the MZ of Young, although a decrease of Cer and HexCer species was observed following chilling in this root zone (Figure 9B). Similar to the EZ, cold stress also caused the increase of phosphoglycerolipids in the MZ of cold-sensitive Wyalkatchem, but not in the MZ of cold-tolerant Young (Figure 10). As shown in Figure 10A, chilling did not cause any significant changes in PC levels, but an increase of two PE and one PG species was observed in the MZ of Wyalkatchem. Among these three species, only PC(18:3_24:1) was increased by more than +2.0-fold. Compared to chilling stress, freezing stress caused even more increases of PC, PE, and PG in the MZ of Wyalkatchem (Figure 10B). A total of 18 PC species were significantly increased due to freezing, but only 12 of them by more than +2.0-fold: PC(18:0_18:3), PC(18:1_18:2), PC(18:1_18:3), PC(18:1_20:1), PC(18:2_20:1), PC(18:3_20:1), PC(18:3_20:2), PC(18:3_22:1), PC(18:2_24:1), PC(18:3_24:0), PC(18:3_24:1), and PC(18:3_26:1). Eighteen PE species were significantly increased in the MZ of Wyalkatchem, with 16 of them by more than +2.0-fold: PE(14:0_18:3), PE(16:0_18:3), PE(18:0_18:3), PE(18:1_18:2), PE(18:1_18:3), PE(18:2_18:3), PE(18:3_18:3), PE(18:2_20:1), PE(18:3_20:0), PE(18:3_20:1), PE(18:2_22:1), PE(18:3_22:0), PE(18:3_22:1), PE(18:2_24:1), PE(18:3_24:0), and PE(18:3_24:1). A total of six PG species were also increased due to freezing in this root zone in Wyalkatchem, and three of them by more than +2.0-fold: PG(16:0_20:1), PG(18:1_18:3), and PG(18:3_18:3) (Figure 10B).

## 3. Discussion

### 3.1. Lipid Composition at the Unstressed Stage Reveal Varietal Differences among the Three Developmental Root Zones in Each Wheat Variety

The DZ is closest to the root tip, and it is made up of actively dividing cells of the root meristem. Of the three root development zones, the EZ is where the newly formed cells increase in length and stimulate root growth. Once the cells have reached their final size at the end of the EZ, they will enter the MZ, which is marked by the beginning of root hair formation, where root cells begin to differentiate into special cell types [49]. In cold-sensitive Wyalkatchem and cold-tolerant Young, it was found that both varieties shared similar patterns of lipid composition at the unstressed stage. Phosphoglycerolipids such as PE, PG, PC, and their lyso species occupied more than 50% of the total lipid composition in the DZ where cells are actively dividing. The percentage gradually decreased across the EZ and was the lowest for the MZ. Conversely, sphingolipids such as HexCer and Cer were the lowest in the DZ for both varieties, then gradually increasing across the EZ and becoming the prevalent category of lipids in the MZ (40% of total measured lipid).

Phosphoglycerolipids, more commonly known as phospholipids, are lipids containing two hydrophobic fatty acyl groups and a hydrophilic polar head group consisting of a phosphate molecule attached to the glycerol group. Plant phospholipids are from the PC (phosphatidylcholine), PE (phosphatidylethanolamine), PG (phosphatidylglycerol), PI (phosphatidylinositol), and PS (phosphatidylserine) classes. Phospholipids can form a lipid bilayer due to their amphiphilic characteristic; they serve as the structural basis of cellular membranes. Amongst the five phospholipid classes, PC and PE are the two most abundant phospholipids that are present in plasma and extraplastidic membranes, while PG lipids are more prevalent in plastidic membranes (e.g., thylakoid membranes) [54]. The highest proportion of phospholipids found in the DZ compared to the other two root zones may be due to the active cell division in this zone, in which large amounts of phospholipids are needed to build new plasma and endo-membranes of dividing cells.

In contrast to the prevalent glycerolipids that have glycerol in their structures, complex sphingolipids are composed of a sphingoid long-chain base (LCB, a kind of amino alcohol) with one amide-linked fatty acyl chain and a polar head group. The predominant sphingoid LCBs in plants are C18 amino alcohols, and the dihydroxy LCB sphinganine (d18:0; dihydrosphingosine) and the trihydroxy LCB 4-hydroxysphinganine (t18:0; phytosphingosine) are commonly found in plant tissues. The LCBs can be N-acylated with fatty acids to form ceramides (Cer). The fatty-acyl chains of plant ceramides are predominantly α-hydroxy fatty acids varying in length from 16 to 26 carbons and are linked to the long-chain base by an amide bond. Ceramides can serve as precursors to more complex sphingolipids such as the monohexosyl ceramide (HexCer) and derivatives of inositolphosphorylceramide (IPC) in higher plants [55,56,57,58]. They can also act as signaling molecules in plants [59]. Glucosylceramide (GlcCer) is the common HexCer found in plants, and the polar head group (glucose) is linked to the C-1 of the N-acyl LCB. GlcCer are components of the plasma membrane and the tonoplast of plant cells [60,61], and are thought to increase membrane stability, decrease membrane permeability, and regulate ion permeability as a consequence of the associated very long (>C20) hydroxyl fatty-acyl chain and the intra- and intermolecular hydrogen bonding between amide and hydroxyl groups of the ceramide moiety [62,63]. The increased hydroxylation of the sphingolipid molecules has also been reported to be associated with increased stability and decreased permeability of membranes [64]. Heavily hydroxylated species of GlcCer present in many plant tissues may contribute to the overall integrity of the plasma membrane and tonoplast, the two membranes most enriched in sphingolipids. These two membranes have critical and variable functions as barriers [57]. Steponkus [65] reported the involvement of sphingolipids to regulate ion permeability and osmotic adaptation of cells to freezing and water deficit. Ceramide and long-chain base metabolites have been shown to regulate cellular processes, including programmed cell death and G protein-mediated guard cell closure [66,67]. In this study, it was found that the proportion of sphingolipids (HexCer and Cer) was the highest in the MZ of both varieties. In the MZ, cells differentiate for various important specialized functions: some cells of the pericycle form lateral roots, epidermal cells may form root hairs, and the Casparian strip will develop between cells of the endodermis [68]. The stability and integrity of those membranes in the MZ need to have a different lipid composition, and the high proportion of sphingolipids which function to regulate membrane permeability and overall integrity of the plasma membrane and tonoplast may relate to this.

### 3.2. Cold Stress Mainly Alters the Glycerolipid and Sphingolipid Compositions of the Root Zones, which May Reveal the Main Differences between the Cold-Sensitive and Cold-Tolerant Varieties

After the wheat seedlings were subjected to six hours of chilling (4 °C) or freezing (−3 °C), it was interesting to observe the increase of phospholipids in every root zone for both wheat varieties. Uemura et al. [69,70] reported that when plants are exposed to low and non-freezing temperatures, the content of phospholipids and degree of fatty acid unsaturation typically increase. These lipid changes are important to enhance membrane fluidity and reduce the propensity of cellular membranes to undergo freezing-induced non-bilayer phase formation, thus maintaining membrane integrity at low temperatures. The significant increase of phospholipids (PC, PE, and PG species) in the cold-sensitive Wyalkatchem but not in Young following chilling and freezing stresses, especially in the EZ and MZ, may indicate the involvement of these lipids in lipid remodeling of the root membranes of the sensitive variety to maintain the fluidity in cold conditions. When comparing the phospholipids between the two wheat varieties, it was observed that fold changes of phospholipids in Young were majority lower compared to Wyalkatchem following chilling and freezing (Appendix A). This again suggests that cold stress may have a stronger impact on the membrane stability of the roots of the sensitive variety compared to the roots of the tolerant variety. The root membrane of the cold-tolerant Young variety may possess an adaptation mechanism to help the variety be more ready for cold stress. The Young variety also showed superior ability to maintain its membrane fluidity upon prolonged cold stress from the studies reported on the flag leaf [42] and spike [43]. In the flag leaf lipidomics study of Wyalkatchem and Young subjected to cold stress [42], only the phospholipid class was analyzed by using LC-triple quadruple-MS, in which the identification and quantification were based on multiple reaction monitoring (MRMs). From the study, it was observed that Young showed a higher unsaturation-to-saturation ratio compared to Wyalkatchem after four days of prolonged cold stress. Both varieties accumulated some saturated and monounsaturated species (e.g., PC34:0, PC34:1, and PC35:1) after one day of cold stress. However, the Young variety successfully decreased the accumulation of these saturated and monounsaturated species (e.g., PC34:0, PC34:1, and PC35:1) after the prolonged cold stress, while Wyalktachem failed to do so. This subsequently led to a higher unsaturation-to-saturation ratio in Young compared to Wyalkatchem in response to prolonged cold stress, indicating that the Young variety may have a superior ability to maintain its membrane fluidity under prolonged cold stress. In the spike lipidomics study [43], Young again showed a higher unsaturation-to-saturation ratio compared to Wyalkatchem after four days (prolonged) of cold stress by having significantly higher amounts of a few polyunsaturated species such as PC36:6, PC38:6, PE36:6, PG36:6, and PI36:6. All the findings gained from the lipidomics study on the phospholipids of root, flag leaf, and spike postulate that the cold-tolerant Young variety may possess an adaptation mechanism to help the variety be more ready for cold stress.

Welti et al. [8] reported that cold exposure induces significant increases in specific lysoPC, lysoPE, and phosphatidic acid (PA) species. Lysophospholipids and PA are produced by phospholipase A and phospholipase D enzymes, which are activated during cold acclimation. Lysophospholipids and PA are minor phospholipids with potential regulatory functions, such as activation of target signaling proteins, regulation of cytoskeletal organization, and regulation of ion channel function. They may play both structural and regulatory roles in plant adaptation and survival during low temperature exposure [6]. PA was not detected in this study, so these results could not be inferred. However, lysophospholipids, such as lysoPC, lysoPE, lysoPG, lysoPI, and lysoPS, were measured in this study and observed to increase only in the EZ of both Wyalkatchem and Young upon freezing stress. This may indicate the involvement of lysophospholipids for the adaptation and survival of the elongating tissues of wheat roots during freezing conditions. Meanwhile, Young has shown higher amounts of lysophospholipids compared to Wyalkatchem in all the three root zones in all three conditions. The higher amounts of lysophospholipids in Young compared to Wyalkatchem may indicate the potential of these lipid classes to play regulatory roles and thus confer greater cold tolerance in Young.

Cold stress also caused a significant increase of several DG species in the DZ but had lesser effects on the EZ of both varieties. The effects were greatest on the MZ of Wyalkatchem (highest numbers of significantly increased lipids), but not on the MZ of Young. The accumulation of DG under low temperature stress has been reported to be due to two main mechanisms: (1) involvement as lipid signaling molecules, and (2) induced during freezing injury causing membrane leakage. For (1), both Xiong et al. [20] and Arisz et al. [21] reported that under freezing conditions, a fast induction of PA occurs through the phosphorylation of phosphoinositide (PtdIns) of the membrane to phosphatidylinositol 4,5-biphosphate, which is then further hydrolyzed by phospholipase C into the second messengers inositol 1,4,5-triphosphate (Ins(1,4,5)-P3) and DG. The latter is then converted to PA by the action of diacylglycerol-kinases (DGK). For (2), Moellering et al. [27] and Lu et al. [71] reported that cellular dehydration increases the concentration of Mg^2+^ and cellular acidity during freezing stress. Mg^2+^ and protons are then transported into the chloroplast outer envelope membrane, which activates the function of SENSITIVE TO FREEZING2 protein (SFR2). SFR2 then converts MGDG into oligogalactolipids and DG. Oligogalactolipids are important for enhancing membrane stability, whereas DG reduces stability. The reason the accumulation of DG is adverse for the membrane is that it can be converted to PA by the DGK enzyme. PA can bind and activate the enzyme NADPH oxidase (RbohD), subsequently increasing the production of reactive oxygen species (ROS) [72]. ROS can play the role of signal transduction molecules during a stress response when their levels are low, but they are harmful when present excessively, leading to the oxidation of membrane lipids and other molecules [73]. PA can also induce the formation of an unstable hexagonal II (HII)-type lipid phase with DG or MGDG, which destabilizes the cell membrane [27,30,74]. To avoid the adverse effects caused by the excessive accumulation of DG and PA, the diacylglycerol transferase (DGAT) enzyme converts DG to (triacylglycerol) TG instead of PA. Evidence has shown that plants that better withstand freezing stress show an up-regulation of DGAT activity and accumulation of TG upon cold stress [75,76]. Upon chilling and freezing stress, both the root zones (especially elongation and maturation zones) of cold-tolerant Young and cold-sensitive Wyalkatchem showed increased DGs as compared to the unstressed state. However, the increases were mostly significant for Wyalkatchem, but not for Young. When the root zones between the two varieties were compared to DGs after being subjected to cold stress, Young showed significantly lower fold changes compared to Wyalkatchem. The lower amount of DG in the cold-tolerant Young and higher amount in the cold-sensitive Wyalkatchem upon cold stress could be due to the expression of the *DGAT* gene, and the activity/efficiency of the DGAT enzyme to convert DG to TG. From the results shown in this study, Wyalkatchem may have a lower expression and activity of the DGAT enzyme to convert DG to TG as compared to Young. However, this is a postulation, and it is thus very important to investigate the DGAT expression in the two varieties upon cold stress, as well as measure the DG and TG contents in the future to help us to gain a better understanding of the cold defense mechanism of tolerance and sensitive varieties. In this study, we modified the parallel reaction monitoring (PRM) setting based on [50]. We tried to include the detection of as many lipid species as possible in our study. Unfortunately, the TGs and oxidative related cardiolipins were not included in our modified PRM. In our future studies on cold stress for more wheat varieties, these two lipid classes will be included.

In contrast to phosphoglycerolipids and DG that were induced upon cold stress, decreases of sphingolipids (Cer and HexCer) in each of the root zones for both wheat varieties were observed. Decreases in the proportion of GlcCer (glucosylceramide, a kind of HexCer) in plants subjected to low temperature stress and during cold acclimation to acquire freezing tolerance has been reported. The proportion of GlcCer in plasma membranes of freezing-tolerant plants is reported to be lower than in freezing-sensitive plants [69,70,77,78]. The reduction of GlcCer composition during cold stress alters the membrane behavior during osmotic excursions and prevents dehydration-induced demixing of membrane lipid components [79,80]. It has also been reported that trihydroxy-LCB having a cis double bond at the ∆8 position (eg. t18:18) are more prevalent in chilling-resistant and freezing-tolerant plants [81,82,83]. Imai et al. [81,82,83] also reported that monounsaturated hydroxy fatty acyl chains (especially h24:1, with a fatty acid chain of 24 carbons, one double bond, and one hydroxyl group attached to one of the carbons—usually the second carbon) are common in freezing-tolerant plants, whereas sensitive plants contain exclusively saturated hydroxy fatty-acyl chains [81,83]. In this study, a decrease in HexCer (i.e., GlcCer) was observed in the three root zones of the cold-tolerant Young but not cold-sensitive Wyalkatchem upon cold stress, where the majority of the significant decreases occurred in the DZ. The lower amounts of HexCer in Young compared to Wyalkatchem upon cold stress may be one of the factors contributing to the cold tolerance of Young. This is a major finding of this study, and a more in-depth study of this class of lipids is justifiable with respect to how lipids are involved in membrane remodeling in wheat following cold stress.

Wyalkatchem also showed different profiles for galactosyl glycerolipids compared to Young. All the galactosyl glycerolipids that have diacylglycerol in their structures, such as DGDG, MGDG, and SQDG, are higher in Wyalkatchem compared to Young in all three root zones at unstressed and cold stress conditions. Conversely, all the galactosyl glycerolipids that have monoacylglycerol in their structures, such as DGMG, MGMG, and SQMG, are higher in Young compared to Wyalkatchem. Galactosyl glycerolipids are mainly abundant in the thylakoid membranes of plastids, where 50% is MGDG, 26% is DGDG, and the remaining are sulfoquinovosyldiacylglycerol lipids (SQDG) and phosphatidylglycerol lipids (PGs) [10]. DGMG, MGMG, and SQMG are scarce in the thylakoid membranes and are believed to be the hydrolysis products of DGDG, MGDG, and SQDG, respectively. For example, MGDG are synthesized in plants by galactosylation of DG in the plastid envelope [84]. MGMG, on the contrary, have been thought to be principally formed by lipase hydrolysis of MGDG [85]. The observation of higher MGMG, DGMG, and SQMG in Young compared to Wyalkatchem is interesting. However, there is still a lack of literature on the roles of those galactosyl monoacylglycerol lipids in plant cold stress responses, especially in roots where plastids are not fully developed—unlike leaves. More attention is paid to galactosyl diacylglycerol lipids in leaves (MGDG and DGDG) in plant cold stress responses [27,86,87]. Moellering et al. [27] have reported that MGDG favors the formation of non-bilayer HII-type structures brought by dehydration during freezing stress. The non-bilayer structures are formed at the interface of apposed membranes and are believed to initiate at the chloroplast envelope membranes during freezing, thus resulting in fusion between bilayers and damage to the membranes. The authors also pointed to the importance of a protein called SENSITIVE TO FREEZING2 (SFR2) that converts MGDG into DGDG, oligogalactolipids, and DG during low temperature stress to reduce the formation of non-bilayer HII-type structures [27]. The ratio of MGDG to DGDG has therefore been studied in plant cold stress response, but with a focus on leaf tissues. Moreover, the potential for the formation of MGMG and DGMG in plants during stress via the acylation of (1) two MGDGs to acylated MGDG and MGMG, or (2) one MGDG plus DGDG to become acylated MGDG and DGMG, has been reported [88]. The observation of increased MGMG and DGMG with decreased MGDG and DGDG during chilling and freezing stresses of the two wheat cultivars in this study could be associated with the acylation response during cold stress. A further analysis of this phenomenon needs to be explored to draw conclusions regarding potential differences in these mechanisms in Wyalkatchem and Young.

## 4. Materials and Methods

### 4.1. Chemicals and Reagents

All chemicals and solvents were purchased from Sigma-Aldrich (Australia) and were either of analytical or mass spectrometry grades. Lipid standards (PC(13:0/13:0), PE(12:0/12:0), PG(12:0/12:0), PS(12:0/12:0), PA(12:0/12:0), Cer(d18:1/12:0), LysoPC(13:0), Sitosterol, and Cholesteryl-ester (9:0)) were purchased from Avanti Polar Lipids (Alabaster, AL, USA).

### 4.2. Lipid Nomenclature, Annotation and Abbreviations

The nomenclature and notation for lipids in this study were referred to LIPID MAPS (http://www.lipidmaps.org/data/classification/LM_classification_exp.php; date accessed 31 December 2020) [50,51]. For example, the nomenclature of PC(16:1_18:2) indicates a PC species with two fatty acyl (FA) chains, 16:1 (16 carbons with one double bond), and 18:2 (18 carbons with two double bonds), but the exact sn-position (sn-1 or sn-2) of the two esterified fatty acyl chains is unknown. The abbreviations for lipids measured in this study were (1) sterols such as ASG: acylated steryl glycoside; SG: steryl glycoside; SE: steryl ester, and ST: steryl derivative; (2) sphingolipids such as Cer: ceramide and HexCer: monohexosyl ceramide; (3) DG: diacylglycerol; (4) galactosyl glycerol lipids such as DGDG: digalactosyl diacylglycerol; DGMG: digalactosyl monoacylglycerol; MGDG: monogalactosyl diacylglycerol; MGMG: monogalactosyl monoacylglycerol; SQDG: sulfoquinovosyl diacylglycerol and SQMG: sulfoquinovosyl monoacylglycerol; (5) phosphoglycerolipids (including their lyso species) such as PC: phosphatidylcholine; PE: phosphatidylethanolamine; PG: phosphatidylglycerol; PI: phosphatidylinositol; and PS: phosphatidylserine; and (6) CL: cardiolipin.

### 4.3. Plant Material and Growth Conditions

The two wheat varieties used in this study, Wyalkatchem (cold-sensitive) and Young (cold-tolerant), were obtained via the National Frost Initiative, Australia (http://www.nvtonline.com.au/frost/; date accessed 31 December 2020). Seeds were sterilized and germinated according to Shelden et al. (2013), with some modifications. Briefly, seeds were surface-sterilized in 70% (*v*/*v*) ethanol for 1 min, followed by 1% (*v*/*v*) sodium hypochlorite (bleach) for 10 min, and rinsed six times with sterile ultrapure water (Millipore, Billerica, MA, USA). The seeds were then imbibed in sterile ultrapure water for 16 hrs with aeration at room temperature in the dark. After that, the seeds were transferred to Whatman^®^ Grade 50 filter papers (Cytiva, Marlborough, MA, USA) and pre-wetted with sterile ultrapure water in round petri dishes (100 mm × 20 mm, Greiner Bio-One, Kremsmünster, Austria; 10 seeds per plate; Appendix A). The seeds were allowed to germinate with the radical orientated downwards by placing the dishes at a 45° angle for 48 hrs at 21 °C in the dark (covered with aluminum foil). Uniform germinated seedlings with the longest seminal root between 1.5 and 2.0 cm (Appendix A) were selected and transferred to square petri dishes (100 mm × 100 mm × 20 mm) containing a nutrient agar medium with a modified Hoagland’s nutrient solution (Genc et al., 2007). Two seedlings were placed on each plate and allowed to grow with the radical orientated downwards by placing the dishes at a 45° angle for another 48 hrs at 21 °C in the dark (Appendix A).

### 4.4. Stress Treatment and Harvest of Root Samples

After 48 h grown at 21 °C in the dark, the seedlings either remained at 21 °C for 6 h in the dark (control treatment), transferred to a cold chamber at 4 °C for 6 h in the dark (chilling stress), or were transferred to a frost chamber at −3 °C for 6 hrs in the dark (freezing stress). A total of 20 plates of seedlings (2 seedlings per plate × 20 plates = 40 seedlings) were treated as one biological replicate (n). There were four biological replicates (n = 4) for each condition (control, chilling, and freezing) (Appendix A). For the harvest of root samples, the apical region of the root was cut into three zones/sections measured from the root tip (Figure 1). From Figure 1, DZ comprised the root cap and cell division zone (cut 1.5 mm from the root tip and labelled as DZ), EZ represented the elongation zone (after cutting 1.5 mm of the DZ, another 1.5 mm was cut and labelled as EZ), and MZ represented the maturation zone (after cutting the EZ, another 2.0 mm was cut and labelled as MZ). The root sections were cut with a pre-chilled scalpel and immediately snap-frozen with liquid nitrogen. The root samples were weighed and recorded (average = 40 mg) prior to being kept at −80 °C for liquid extraction.

### 4.5. Lipid Extraction

The extraction of lipids was carried out as described by [89], with some modifications. Frozen root sections were transferred into Cryomill tubes (Precellys 24, Bertin Technologies, Rockville, MD, USA). Subsequently, 500 µL of chloroform:methanol:water (1:3:1, *v*/*v*/*v*) and 5 µL of 1% butylated hydroxytoluene (BHT) was added to the samples, followed by vortexing for 30 s and homogenization at −10 °C using a Cryomill (3 × 45 s at 6100 rpm). The samples were then extracted for 30 min at 25 °C in a thermomixer at 1400 rpm, and subsequently centrifuged for 10 min at 4 °C at 15,000 rpm. The supernatants were transferred into new tubes. 500 µL of the same extraction solvents and BHT were added into the remaining pellets for second extraction. The supernatants were collected and combined, vortexed for 30 secs, and re-centrifuged for 10 min at 15,000 rpm. A 200 µL aliquot of 0.1 M hydrochloric acid (HCl) was then added in, followed by vortexing and centrifugation (10 min, 15,000 rpm). The bottom layers of the supernatants (100 µL for each sample) were carefully transferred to new tubes and dried using SpeedVac at 40 °C. The dried samples were kept at −80 °C prior to lipid analysis.

### 4.6. LC-ESI-MS/MS Analysis

The samples were analyzed using a LC-ESI-QqTOF system as described in [50], with some modifications. The samples were first re-constituted with 150 µL of isopropanol:methanol:water (4:4:1, *v*/*v*/*v*). Internal standard mixtures of PC(13:0/13:0) and Cer(d18:1/12:0) were added into each sample in a 1 µM final concentration prior to LC-MS analysis. An extra sample called the pooled biological quality control (PBQC) sample was then produced by collecting 30 µL from each sample. Samples (10 uL each) were injected to an Agilent 1290 HPLC system (Santa Clara, CA, USA) coupled to a SCIEX TripleTOFTM 6600 QqTOF mass spectrometer (Framingham, MA, USA). Separation of lipid species was carried out using an Agilent Poroshell EC-C18 column (100 mm × 2.1 mm, 2.7 µm), at a flow rate of 0.4 mL/min, and run in a linear gradient based on two mobile phases. Mobile phase A consisted of methanol and 20 mM ammonium acetate in a 3:7 ratio (*v*/*v*), and mobile phase B consisted of 2-propanol:methanol:20 mM ammonium acetate in a 6:3:1 ratio (*v*/*v*/*v*). The gradient started with 65% B for 2 min, increased to 100% B over 8 min, followed by 100% B for 6 min, and then re-equilibrated to 65% B in 2 min. The PBQC sample was injected after every ten samples. During the run, the instrument was calibrated automatically every 20 samples to maintain the mass resolution of MS1 spectra at ~35,000 FWHM (full width at half maximum) and mass accuracy below 5 ppm, resolution of MS/MS spectra at ~20,000 FWHM, and mass accuracy below 10 ppm. All the ESI and parallel reaction monitoring (PRM) parameters were modified and set according to [50], except the collision energies (CEs). Lipids such as ASG, Cer, HexCer, DG, SG, SE, and ST were analyzed with a positive ion mode at CE −40 V. Meanwhile, lipids such as DGDG, DGMG, MGDG, MGMG, SQDG, SQMG, PC, PE, PG, PI, PS, and CL were analyzed with a negative ion mode at a CE +60 V.

### 4.7. Data Processing

Each of the detected lipids using PRM transitions (MS/MS spectra) was identified using an in-house generated lipid database for barley [50], and the peak area of the identified lipid was based on extracted ion count chromatogram (EICC) for one or multiple fragment ions analyzed using MultiQuant software (SCIEX, Version 3.0.2, Framingham, MA, USA). For phosphoglycerolipids, galactosyl glycerolipids, as well as CLs detected in a negative ion mode, the peak area of all negative charged FA fragments was summed. For DGs detected in a positive ion mode, the total peak area of all fragments resulting from neutral loss of an FA chain was used. For HexCer and Cer species detected in a positive ion mode, the sum of the peak area of positively charged long chain base (LCB) and its dehydrates from up to three dehydration processes were used. For sterols detected in a positive ion mode, the dehydrated sterol backbone was the only fragment chosen. Peak picking for fragment ions was set to a 100 ppm width. Integration settings were as follows: Noise percentage = 40%, and Gaussian smooth width = 2 points. The peak area of each of the lipid species was then normalized to the intensity/responses of internal standards and sample weight. The responses of the PBQC samples for each lipid species were used to calculate the coefficient of variation (CoV). Only lipid species with CoV less than 20% were selected for further data analysis.

### 4.8. Statistical Analysis

Pairwise comparisons were performed first in each root zone of each variety under control versus chilling or control versus freezing stress. Then, pairwise comparisons were performed between the two varieties, Young and Wyalkatchem, for each root zone in control, chilling, and freezing conditions. The statistical significance of differences observed between samples was evaluated using the Student’s *t*-test using Excel with a false discovery rate (FDR)-adjusted *p*-value of 0.05 as a cut-off [90]. The pairwise comparisons of the data were presented as log2-transformed of the fold change values and plotted using Graph Pad Prism 7.0 software (GraphPad Software, La Jolla, CA, USA). The stacked bars were plotted using the same Graph Pad software and BioRender software (https://biorender.com/). Multivariate statistics were performed in R statistical programming language using modified functions from the R package RandoDiStats (https://github.com/MSeidelFed/RandodiStats package) [91].

## 5. Conclusions

Using the modified lipidomics approach from Yu et al. [50,51], a complex lipidome analysis of the developmental root zones of two wheat varieties with contrasting cold tolerances have been conducted in this study. A total of 273 lipid species derived from 21 lipid classes and three categories of lipids has been analyzed. Chilling and freezing have effects mainly on the composition of phosphoglycerolipids, glycerolipids, and sphingolipids in the root zones, revealing the main differences between the cold-sensitive Wyalkatchem and cold-tolerance varieties. Together with the outcomes gained from the studies of flag leaves and spikes [42,43], the outcomes gained from this root study may help us to better understand the mechanisms of wheat response to cold, thus contributing to wheat breeding for cold-tolerant varieties in the future.

## Figures and Tables

**Figure 1 plants-11-01364-f001:**
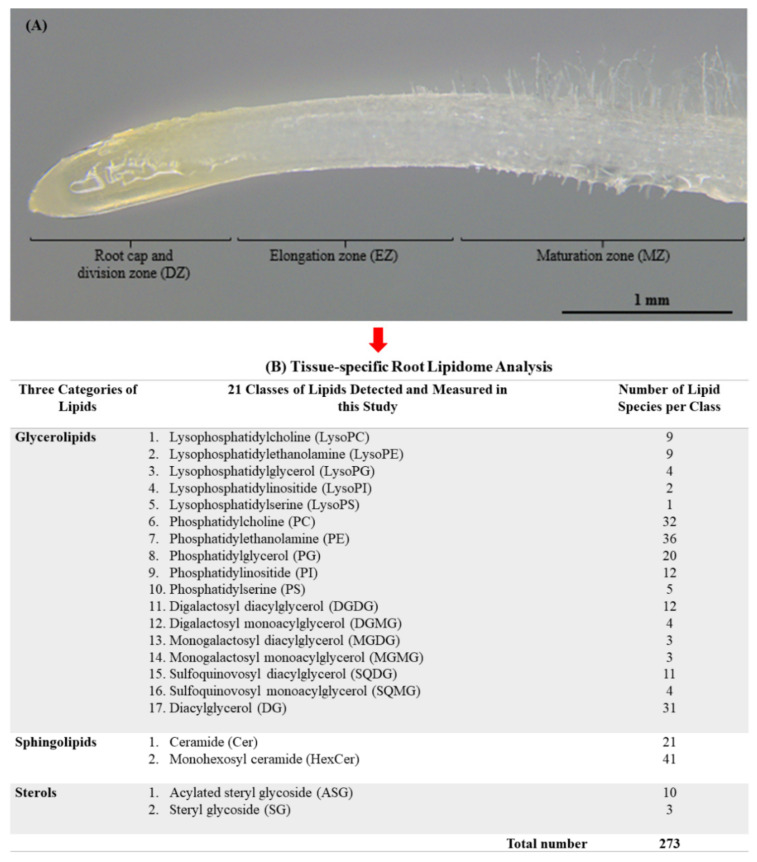
Tissue-specific root lipidome of Young (cold-tolerant) and Wyalkatchem (cold-sensitive) wheat varieties in this study. (**A**) The three developmental root zones of a Wyalkatchem’s young seedling. DZ = root cap and division zone, EZ = elongation zone, MZ = maturation zone. (**B**) A total of 273 lipid species derived from 21 lipid classes and three lipid categories (glycerolipids, sphingolipids and sterols) analyzed in this study.

**Figure 2 plants-11-01364-f002:**
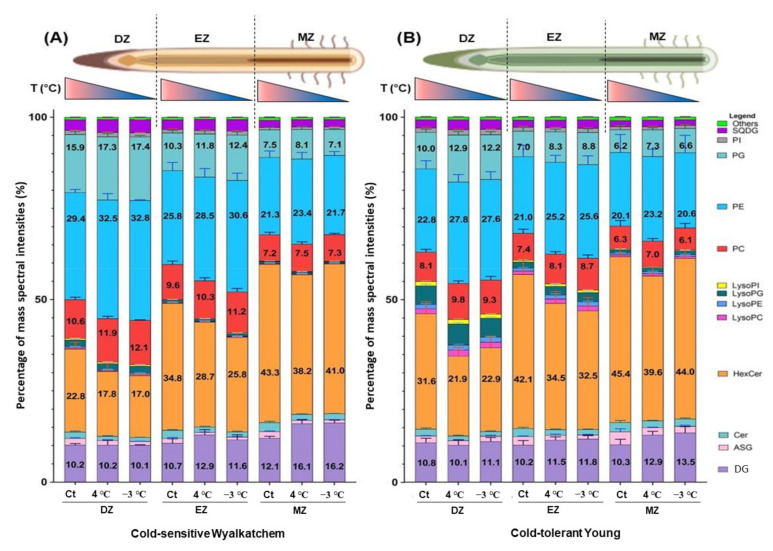
The distribution of the 21 measured lipid classes (in percentage of mass spectral intensity, %) in each specific root zone of (**A**) cold-sensitive Wyalkatchem and (**B**) cold-tolerant Young at the unstressed stage (21 °C) and after subjected to chilling (4 °C) and freezing (−3 °C) stress. DZ = division zone, EZ = elongation zone, MZ = maturation zone, Ct = control (21 °C). Others (light green colour) in the legend represent lipid classes PS, LysoPS, DGDG, DGMG, MGDG, MGMG, SQMG, and SG that in total occupied less than or equal to 1% of the distribution.

**Figure 3 plants-11-01364-f003:**
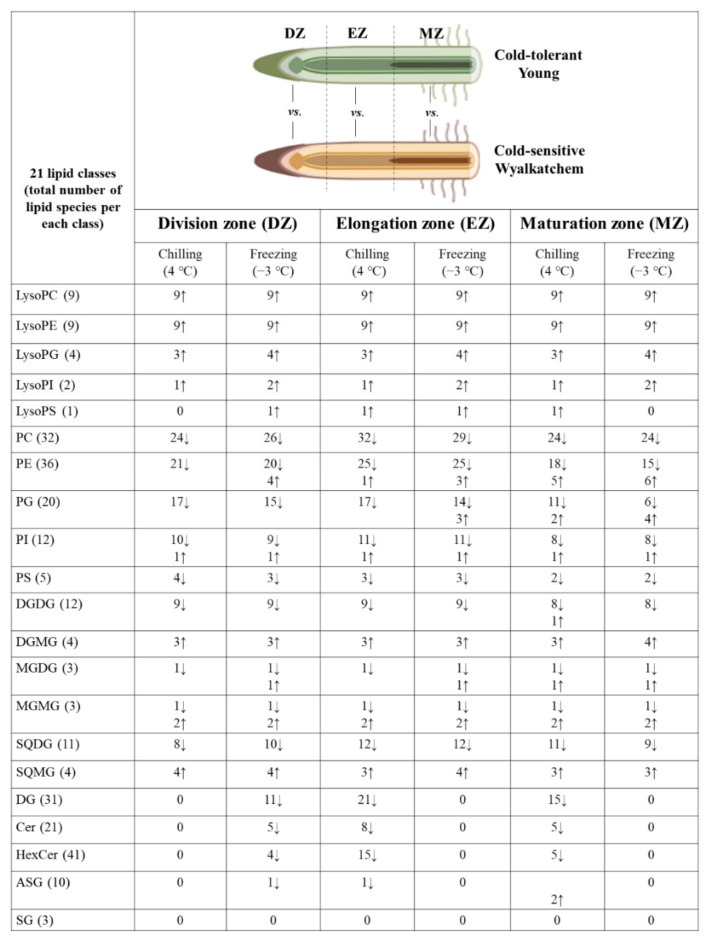
Varietal differences of the lipid profiles of three developmental root zones between Young and Wyalkatchem following chilling and freezing stress treatment. (↑ significantly higher; ↓ significantly lower).

**Figure 4 plants-11-01364-f004:**
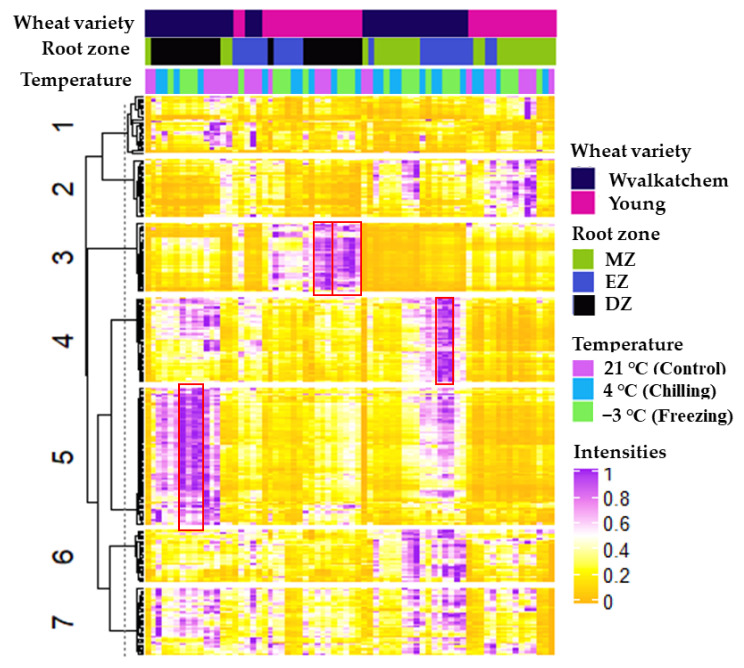
Bootstrapped hierarchical clustering analysis (HCA) of lipid autoscaled intensities from all the root samples (control and subjected to cold stress for three developmental root zones) of cold-sensitive Wyalkatchem and cold-tolerant Young varieties. Rows were clustered using K-means with K equal to 7, and with n clustering reiterations (*n* = 1000). R packages ComplexHeatmap and pvclust were used to do and extract the heatmap image [52,53].

**Figure 5 plants-11-01364-f005:**
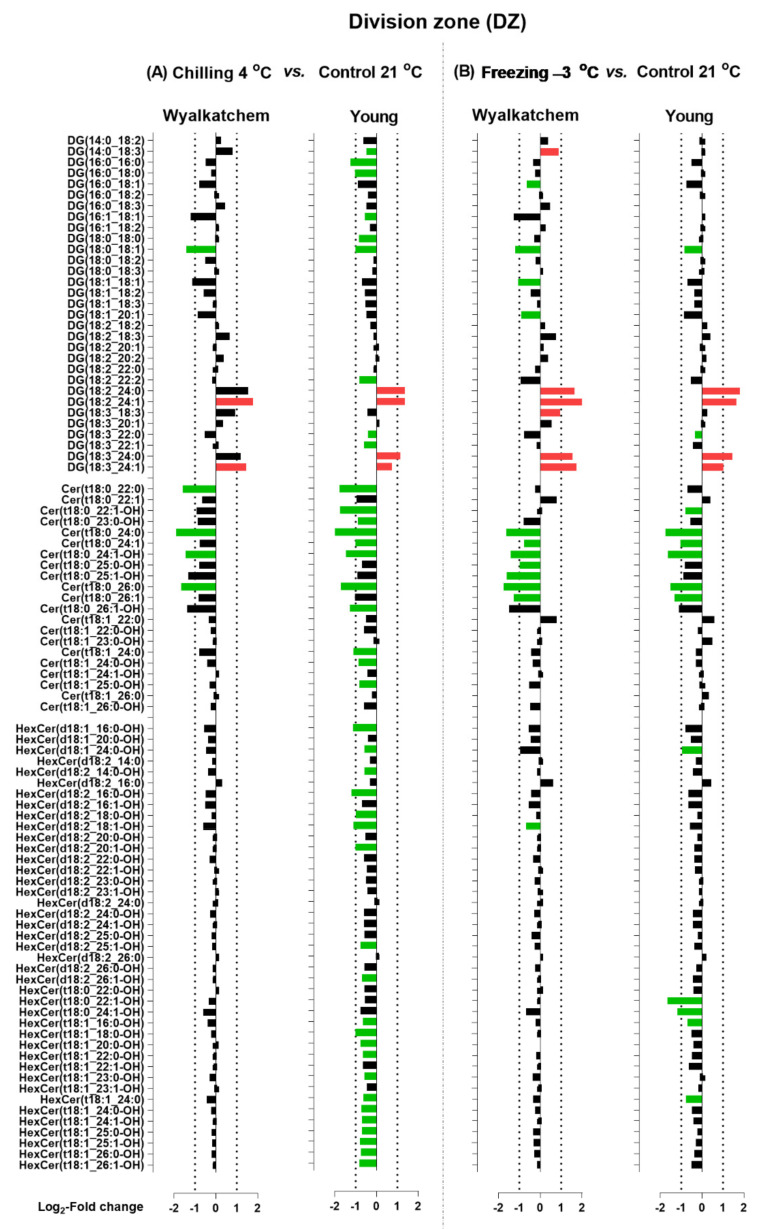
Log_2_-fold changes of DG, Cer, and HexCer species in the division zone (DZ) of cold-sensitive Wyalkatchem and cold-tolerant Young after exposure to (**A**) chilling and (**B**) freezing stress. Statistical significance was determined using an FDR-adjusted *p*-value of 0.05 as the cut-off. Green bar = significant decrease; Red bar = significant increase. Black bar represents lipid species that has no significant change after cold stress as compared to control. *n* = 4.

**Figure 6 plants-11-01364-f006:**
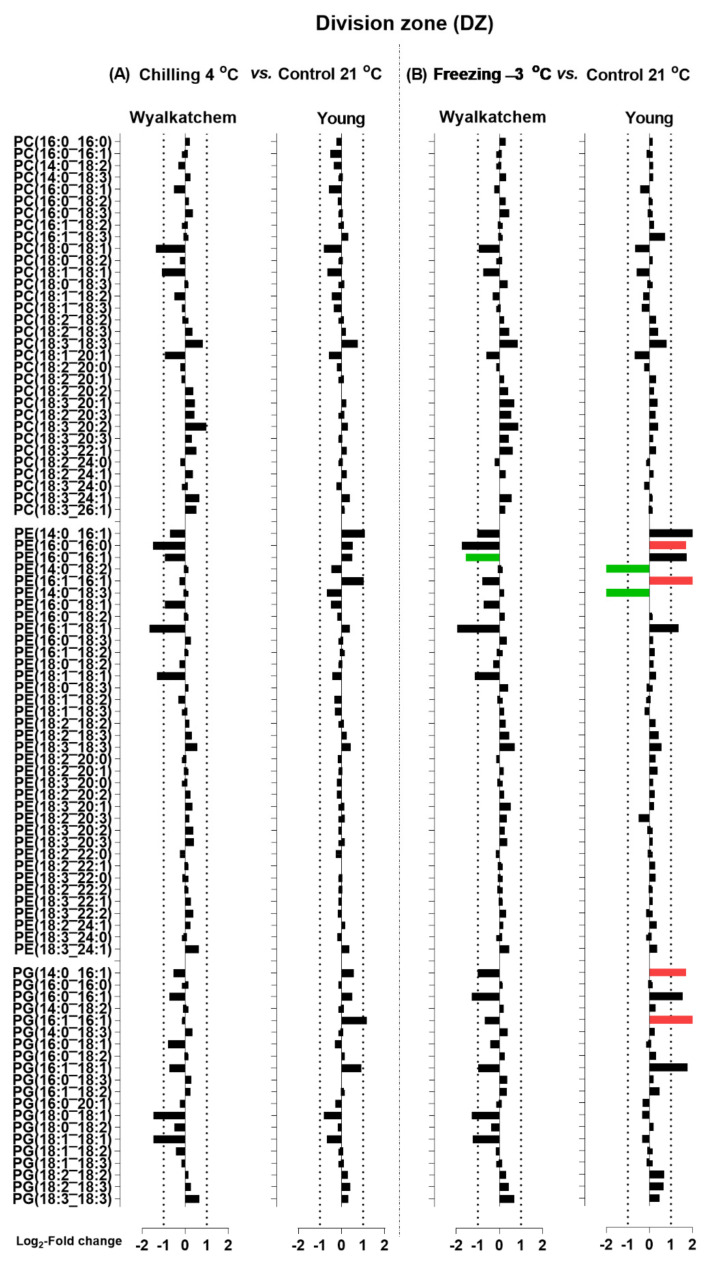
Log_2_-fold changes of PC, PE and PG species in the division zone (DZ) of the cold-sensitive Wyalkatchem and cold-tolerant Young after exposed to (**A**) chilling and (**B**) freezing stresses. Statistical significance was determined using an FDR-adjusted *p*-value of 0.05 as the cut-off. Green bar = significant decrease; Red bar = significant increase. Black bar represents lipid species that has no significant change after cold stress as compared to control. n = 4.

**Figure 7 plants-11-01364-f007:**
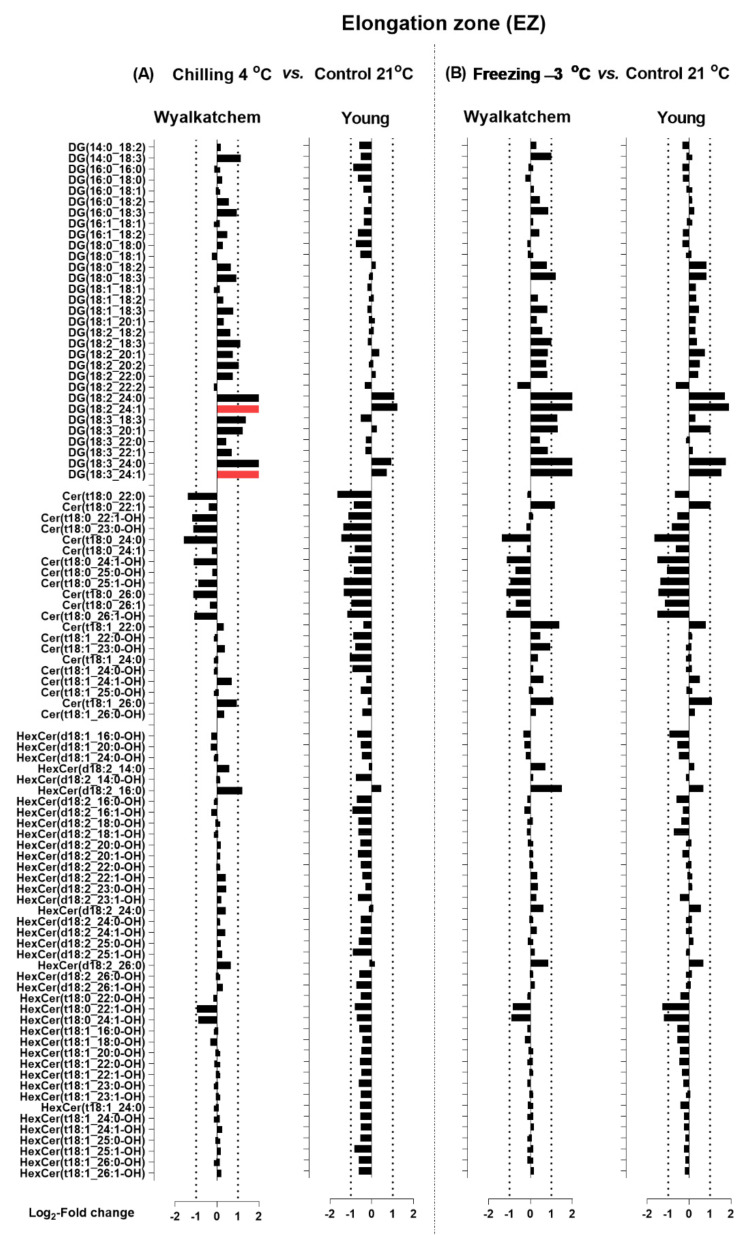
Log_2_-fold changes of DG, Cer and HexCer species in the elongation zone (EZ) of the cold-sensitive Wyalkatchem and cold-tolerant Young after exposed to (**A**) chilling and (**B**) freezing stresses. Statistical significance was determined using an FDR-adjusted *p*-value of 0.05 as the cut-off. Green bar = significant decrease; Red bar = significant increase. Black bar represents lipid species that has no significant change after cold stress as compared to control. n = 4.

**Figure 8 plants-11-01364-f008:**
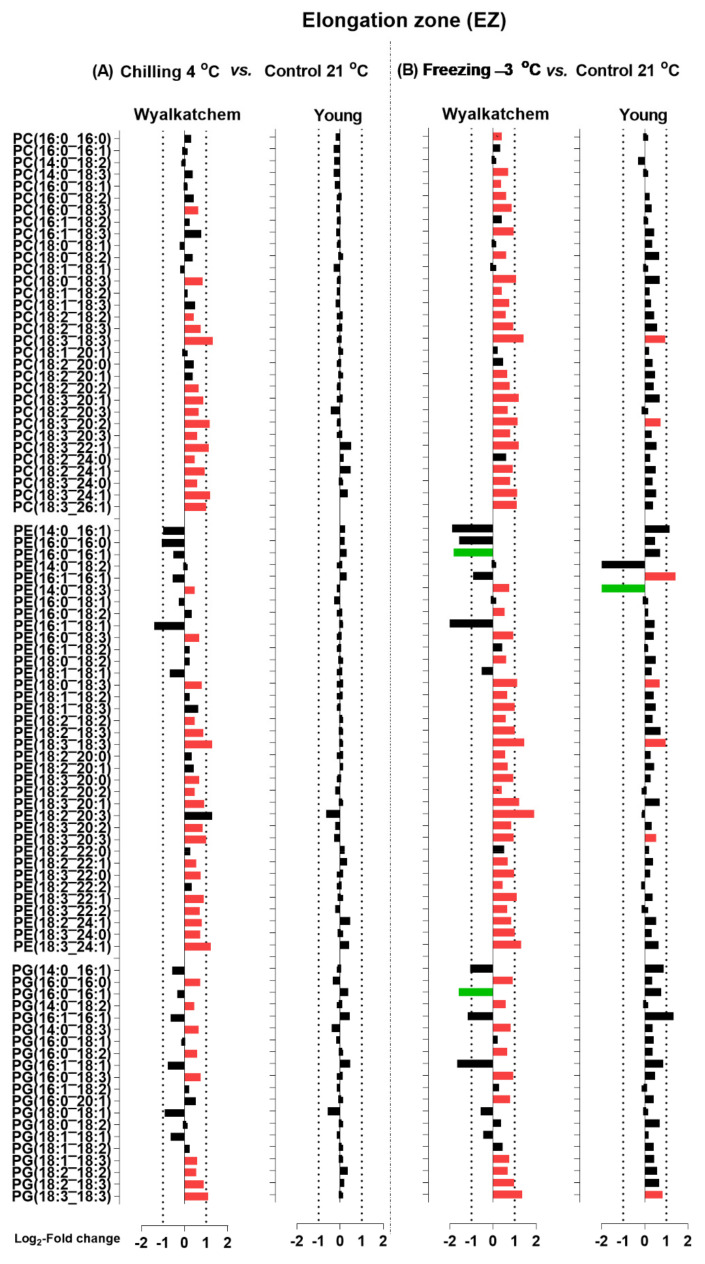
Log_2_-fold changes of PC, PE and PG species in the elongation zone (EZ) of the cold-sensitive Wyalkatchem and cold-tolerant Young after exposed to (**A**) chilling and (**B**) freezing stresses. Statistical significance was determined using an FDR-adjusted *p*-value of 0.05 as the cut-off. Green bar = significant decrease; Red bar = significant increase. Black bar represents lipid species that has no significant change after cold stress as compared to control. n = 4.

**Figure 9 plants-11-01364-f009:**
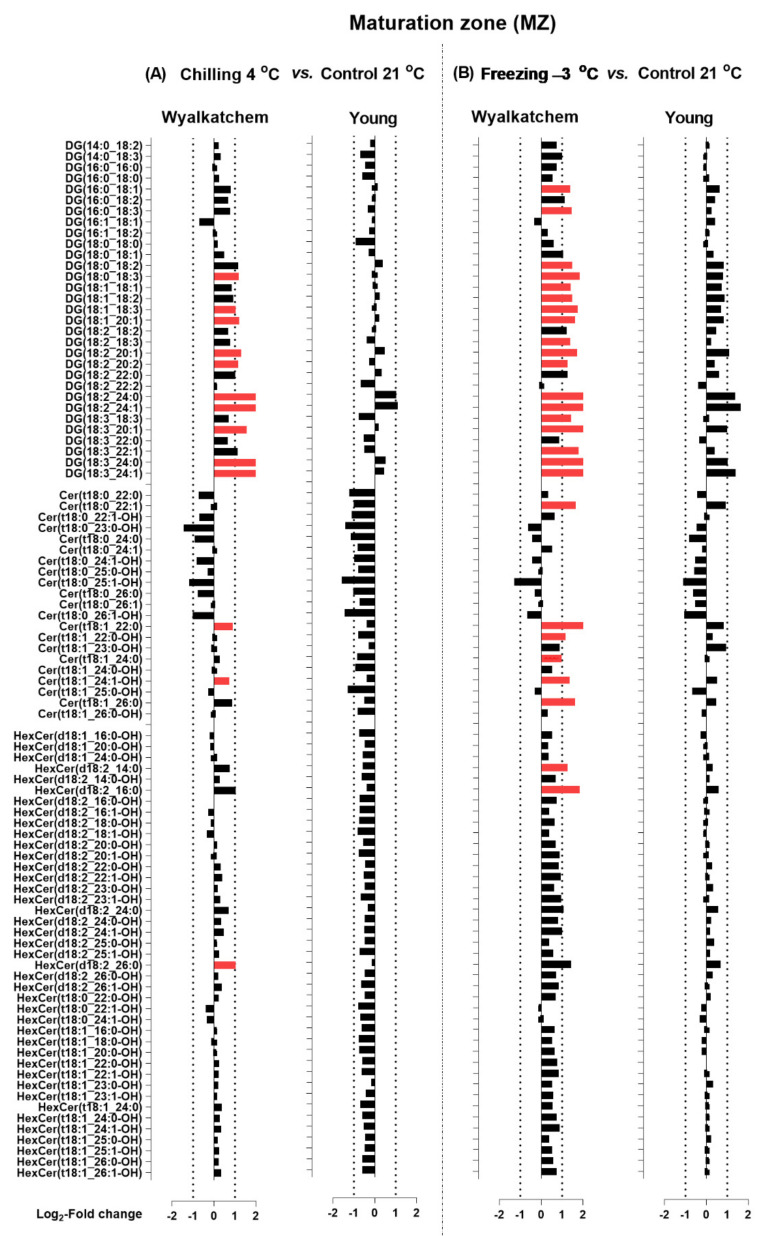
Log_2_-fold changes of DG, Cer and HexCer species in the maturation zone (MZ) of the cold-sensitive Wyalkatchem and cold-tolerant Young after exposed to (**A**) chilling and (**B**) freezing stresses. Statistical significance was determined using an FDR-adjusted *p*-value of 0.05 as the cut-off. Green bar = significant decrease; Red bar = significant increase. Black bar represents lipid species that has no significant change after cold stress as compared to control. n = 4.

**Figure 10 plants-11-01364-f010:**
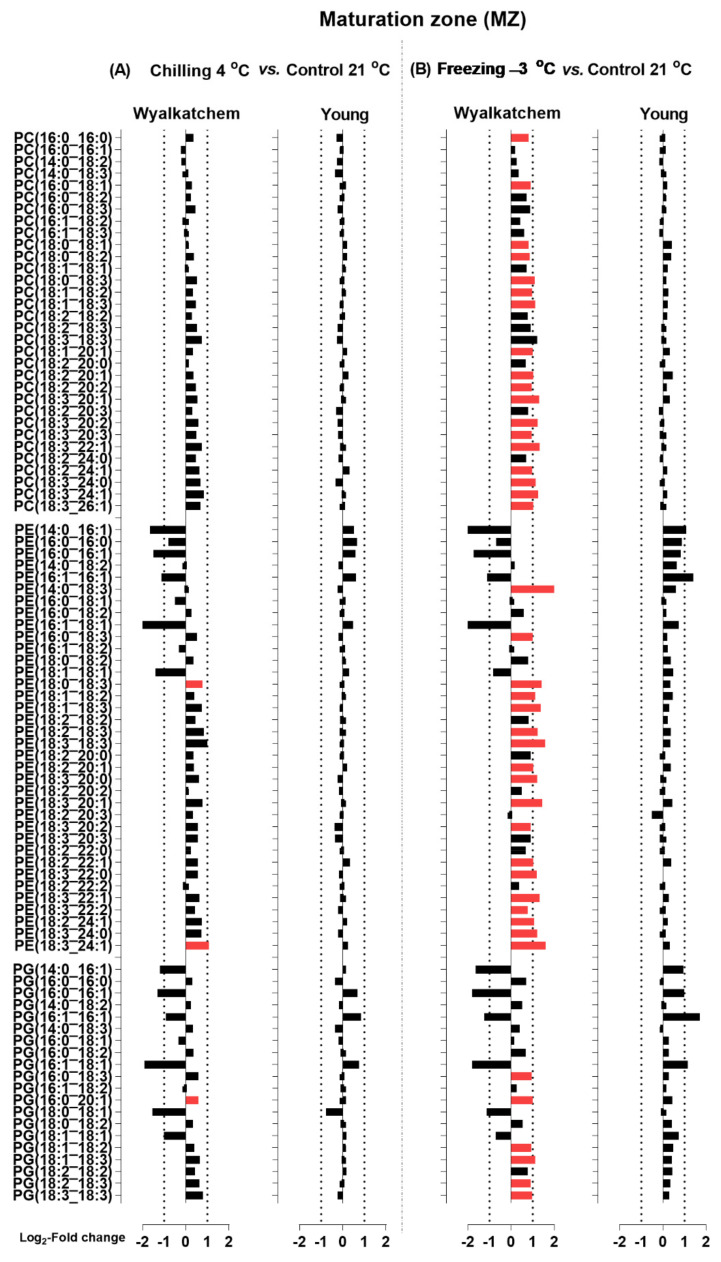
Log_2_-fold changes of PC, PE and PG species in the maturation zone (MZ) of the cold-sensitive Wyalkatchem and cold-tolerant Young after exposed to (**A**) chilling and (**B**) freezing stresses. Statistical significance was determined using an FDR-adjusted *p*-value of 0.05 as the cut-off. Green bar = significant decrease; Red bar = significant increase. Black bar represents lipid species that has no significant change after cold stress as compared to control. n = 4.

## Data Availability

Not applicable.

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
