# Peer review of "The Effect of Cold Stress on the Root-Specific Lipidome of Two Wheat Varieties with Contrasting Cold Tolerance"

_plants, 2022, doi:10.3390/plants11101364_

Round 1

Reviewer 1 Report

Comments: Cheong et al presented on the understanding of lipidome in two wheat varieties where one is cold sensitive (Wyalkatche) and other as cold resistant (Young) one. They elucidated the complete lipid profiling that includes 21 different lipid classes and 271 types of lipids for different root parts and at different temperature stress conditions. The correlation of fold changes in the type of lipids observed in Young and Wyalkatchem varieties evidently shown a decrease of the phospholipids (PC and PE) and a slight increase in lysophospholipids. However, the decrease in DAG is not consistent under chilling and freezing conditions as shown in Table 1. In contrast, a significant fold change was observed for DAG in both varieties in division zone and elongation zone of root tips. The lipidome of normal root tips and its alterations with cold sensitive and cold tolerant varieties shall be discussed if any.

As mentioned in the paper “Arisz et al. [73] reported that plants that better withstand freezing stress show up-regulation of DGAT activity and accumulation of TAG upon cold stress”.  While there is no identification or quantification of TAGs in this study. The change in flux of decrease in phospholipids to neutral lipids (i.e PA to DAG) and understanding the unsaturated lipids of important classes shall be discussed in this manuscript more explicitly. Also, comparison of how the stress conditions in the leaf and spike lipidomes of previous study with that of root tips of current study has to be elaborated in the discussion section for the benefit of readers.

A minor correction: What does black bar represents in Figures 4 and 5 where Log2-fold changes of lipids is represented. Kindly add a note about it.

Reviewer 2 Report

The whole research is relatively systematic, and the results are really credible. 

This study showed the changes of lipid profiles in the different developmental zones of roots of seedlings of two wheat varieties with contrasting cold tolerance exposed to chilling and freezing temperatures, which will fill the research gap of the effects of cold stress on the root lipidome and tissue-specific responses of cold stress wheat roots. Thus, this study is interesting and original. However, before it can be published, I still have 2 suggestions for the paper.

  1. Page 1 line 33, is there a redundant d or missed a word?
  2. In the discussion part, please make an in-depth discussion on the research progress of lipomics related to wheat cold stress. If possible, you can also make a comparison with other crops. In this way, readers can have a comprehensive understanding of the current problems.
